# Ammonia-Oxidizing Bacterial Communities in Tilapia Pond Systems and the Influencing Factors

**Limin Fan \*, Liping Qiu, Gengdong Hu, Chao Song, Shunlong Meng \* , Dandan Li and Jiazhang Chen \***

Freshwater Fisheries Research Center, Chinese Academy of Fishery Sciences, Scientific Observing and Experimental Station of Fishery Resources and Environment in the Lower Reaches of the Yangtze River, Wuxi 214081, China; qiulp@ffrc.cn (L.Q.); hugd@ffrc.cn (G.H.); songc@ffrc.cn (C.S.); lidd@ffrc.cn (D.L.)
\* Correspondence: fanlm@ffrc.cn (L.F.); mengsl@ffrc.cn (S.M.); ffrcchen@hotmail.com (J.C.)

**Abstract:** This study investigated ammonia-oxidizing bacterial communities in water and surface sediments of three tilapia ponds and their relationship with differences in the ponds, monthly variations in the water, and the physico-chemical parameters. Samples were collected from ponds with different stocking densities, after which DNA was extracted, 16S rRNA genes were amplified, the Illumina high-throughput sequencing was performed, and then the Silva and FunGene databases were used to investigate the ammonia-oxidizing bacterial communities. In total, 308,488 valid reads (144,931 in water and 163,517 in sediment) and 240 operational taxonomic units (207 in water and 225 in sediment) were obtained. Further analysis showed that the five genera of *Nitrosospira*, *Nitrosococcus*, *Nitrosomonas*, *Proteobacteria_unclassified*, and *Nitrosomonadaceae_unclassified* were distributed not only in the water, but also in surface sediments of all three ponds. Further, not only the abundance of these five genera, but also their diversities were affected by monthly variations in the water and by sediment differences among the ponds. Moreover, the total nitrogen (TN), nitrate, total phosphorus (TP), and sulphate were the main factors influencing the ammonia-oxidizing bacterial communities in the water, whereas TP was the main influencing factor in the sediments. Moreover, the parameter changes, especially those caused by differences in the ponds, were closely related to the cultivation management (stocking density and feed coefficients).

**Keywords:** tilapia pond; ammonia-oxidizing bacterial communities; stocking density; feed coefficient

## 1. Introduction

Nitrification is one of the crucial steps in the nitrogen cycle in water, where ammonia or ammonium is biologically oxidized to nitrite followed by oxidation of the nitrite to nitrate. As the first step in this process, the oxidation of ammonia to nitrite is carried out by ammonia-oxidizing microorganisms and is usually the rate-limiting step of nitrification [1]; further, it is also closely related to the accumulation of ammonia in systems. In aquaculture systems, due to the constant feeds input, uneaten feeds and feces accumulate, and ammonia is released into the system continuously.

Ammonia is toxic to the culture organisms, and also contributes to the eutrophication of the water [2]. Thus, the removal or the transformation of ammonia from the aquaculture system is important for aquaculture practices. Further, investigation on microorganisms that perform ammonia oxidization is a paramount research focus. Ammonia-oxidizing bacteria, which are the important ammonia removal microorganisms in aquatic ecosystem, mainly consist of two monophyletic groups: beta- and gamma-proteobacteria [3]. The communities have been reported to respond to many physiological or ecological variations of water bodies, such as the water's pH [4] and soluble reactive phosphorus levels [5], and the different culture species of aquaculture ponds [5–8]. Tilapia, an economically and globally important aquaculture food commodity, is considered one of the most promising farmed fish of this century [9]. It is mainly cultivated in freshwater ponds, and could have



significant effects on pond ecosystems since it has strong digestive ability [10] and fast growth rates. Further, the faster accumulation of uneaten feeds and feces may lead to the faster release of ammonia into the system. In addition, due to the habit of nest-digging, tilapia cultures could accelerate the exchange of microbes between the surface sediments and water. Thus, we hypothesized that the ammonia-oxidizing bacterial communities in the water and sediment of tilapia pond ecosystems may respond to cultures of tilapia in different ways and present specific relationships with some physio–chemical parameters. The knowledge about this topic is of significance, as it will help to find ways of reducing ammonia in water. However, studies on the community structures of ammonia-oxidizing bacteria in tilapia aquaculture ponds and their influencing factors have not yet been conducted or reported. Based on our previous research on bacterial communities in water [11] and surface sediments of the same tilapia ponds [12], the same datasets were assigned to the FunGene database to analyze the ammonia-oxidizing bacterial communities in both water and surface sediments. Thus, research on this topic not only supports tilapia pond cultures, but is also an important supplement to the research on the ecology of ammonia-oxidizing bacteria.

In this study, we determined the composition and bacterial species diversity of ammonia-oxidizing bacterial assemblages in both water and surface sediments of tilapia aquaculture ponds with different stocking densities during the whole cultivation season. Our primary focus was to gain an overall understanding of the ammonia-oxidizing bacterial community structures and carry out a preliminary investigation on whether the environment affects the bacteria.

## 2. Materials and Methods

### 2.1. Sample Collection

Three target tilapia ponds for sample collection were situated at an aquaculture farm located in Yixing (31°27′44.2″ N 119°51′2.6″ E) of Jiangsu, China. The area of each pond was approximately equal to one another, at 1330 m$^2$. Tilapia (genetically improved farmed tilapia—GIFT) (*Oreochromis niloticus*) were farmed from fry between 23 May and 15 October 2014. The stocking densities (STD) were 2400 fish/pond (pond 1, the low pond), 3000 fish/pond (pond 2, the medium pond), and 3600 fish/pond (pond 3, the high pond), respectively. The fish were fed after the first month with given formula feed (widely used in China, the composition is as follows: crude protein $\geq$ 26%, crude fat $\geq$ 4%, crude fibre $\geq$ 15%, crude ash $\leq$ 18%, water content $\leq$ 12%, total phosphorus $\geq$ 1%, and lysine $\geq$ 1.2%). Further, the feeding frequency was four times a day (about 08:00 a.m., 11:00 a.m., 14:00 p.m., and 16:00 p.m.). During the whole period of production, there were no changes in the water and no use of fishery drugs. At the end of the farming, the feed coefficients (FC) were 1.51 (the low pond), 1.21 (the medium pond), and 1.26 (the high pond) for each of the three ponds, respectively. The calculation of the feed coefficient was performed using Formula (1).

$$\text{Feed coefficient} = \text{the whole weight of bait put in the pond}/\text{the whole weight of the fish harvested from the pond} \quad (1)$$

Samples from the water and sediment were collected from the three ponds on 30 May, 29 July, 26 September, and 10 October 2014. The tools for collecting the water and sediment samples were plexiglass water samples and a gravity-type cylindrical sampler, respectively. After collection, the tubes containing sediment samples were transported in liquid nitrogen and then stored at $-80\,°C$ until further analysis. A total of 200 mL of water was filtered with a hydrophilic polyethersulphone membrane (0.22 μm for pore size, Jinteng Experimental Equipment Co., Ltd., Tianjin, China) to obtain the target microbes for further analysis. Finally, 12 sediment samples per pond were obtained. The samples from the low pond collected in May were named MS11, MS12, and MS13, respectively, where MS represents May sediment samples, 1 represents pond 1 (the low pond), and the last number represents the three iterations, respectively. Similar naming rules were used for the July sediment

samples (JS), September samples (SS), and October samples (OS). The other ponds were also labelled using this format. The water samples were named with the same scheme.

### 2.2. DNA Extraction, Bacterial 16S rRNA Gene Amplification, and High-Throughput Sequencing Analysis

The DNA in the sediment and in water was extracted using the PowerSoil DNA Isolation Kit (MO BIO, San Diego, CA, USA) and PowerWater DNA Isolation Kit (MO BIO, San Diego, CA, USA), respectively, according to the instructions.

The 515F 5′-barcode-GTGCCAGCMGCCGCGG-3′ and 907R 5′-CCGTCAATTCMTTT RAGTTT-3′ primers were used to amplify the $V_3$–$V_4$ hypervariable region of the bacterial 16S ribosomal RNA gene [13]. The barcode used here is an eight-base sequence unique to each of the samples. The procedure of the PCR amplifications, the purifying method of the amplification products, and the paired-end sequencing method were all the same as in our previous study (Fan et al., 2016). The raw data for the water and sediment samples were all deposited in the NCBI SRA database. The accession numbers are SRP108448 (water) and SRP081260 (sediment).

### 2.3. Bioinformatics and Statistical Analysis

Using the Trimmomatic software [14] and FLASH [15], the acquired raw sequences were demultiplexed and quality-filtered, according to the criteria and main steps of our previous study [12]. The Usearch (Version 7.1) software [16] was used to cluster the operational taxonomic units (OTUs) with 97% similarity cut-off to remove the Chimera and Singletons.

As mentioned above, all sequences were from the same datasets of our previous study. To investigate the ammonia-oxidizing bacterial communities, all these sequences, not only from the surface water, but also from sediments of the tilapia ponds, were assigned to taxa by analyzing the Ribosomal Database Project classifier against the Silva (SSU123) 16S rRNA followed by the FunGene databases with a confidence threshold of 70%. The bacterial community species richness (the ACE index) and diversity (the Shannon index) values were all calculated using the MOTHUR (Version 1.34.3) software [17]. Nonmetric multidimensional scaling (NMDS) [18] and environmental interpretation were conducted using R software. The analysis of similarities (ANOSIM) between ammonia-oxidizing bacterial communities in water and sediments was performed using the PRIMER (Version 5.0) software. The IBM SPSS Statistics 20.0 software was used for statistical analysis, and the least-significant difference (LSD) method was used to perform the multiple comparisons. A $p$ value of <0.05 was accepted as being statistically significant.

### 2.4. Determination of Physico-Chemical Parameters

The concentrations of total phosphorous (TP), total nitrogen (TN), ammonia, nitrite, sulphate, nitrate, total organic carbon (TOC), pH, dissolved oxygen (DO) and Secchi disc depth (SD) in water, and the total phosphorous (sTP), total nitrogen (sTN) and organic matter (OM) in sediment were determined according to the methods in our previous research [19]. They are as follows: TN and TP in water and in sediment were measured by using the alkaline potassium persulfate oxidation method (UV spectrophotometry) and the Kjeldhal method, respectively. The concentrations of nitrite, nitrate and sulfate in water were measured with a DIONEXICS 3000 Ion Chromatograph. A Nessler colorimetric assay was used to measure the concentration of ammonia in water. The TOC in water was assessed using a GE Sievers Innov Ox LaboratoryTOC Analyzer. The dissolved oxygen meter and the oxidation reduction potentiometer were used to measure the values of DO and ORP. The SD in water was measured with the Secchi disc. Meanwhile, the OM content in sediment was determined using the oxidation method.

## 3. Results

According to the comparisons to sequences from the FunGene database, a total of 240 operational taxonomic units (OTUs) (225 in sediment and 207 in water) and 308,448 reads (163,517 in sediment and 144,931 in water) that could be designated as ammonia-oxidizing bacteria were obtained from all samples. The Venn diagram (Figure S1) shows that the shared OTU number between water and sediment samples was 193, accounting for 80.4% of the total. Meanwhile, the unique ones in the water and sediment samples were 15 and 33, accounting for 6.3% and 13.7% of the total, respectively. Thus, there were some differences between the water and sediment samples regarding their ammonia-oxidizing bacterial community compositions. The analysis of similarities (ANOSIM) between ammonia-oxidizing bacterial communities in the water and sediments showed that the differences between water and surface sediments was significant ($p < 0.05$).

### 3.1. Ammonia-Oxidizing Bacterial Communities and Their Influencing Factors

All the OTUs obtained from all samples could be assigned to five genera, *Nitrosospira*, *Nitrosococcus*, *Nitrosomonas*, *Proteobacteria_unclassified*, and *Nitrosomonadaceae_unclassified* (Figure 1), among which *Nitrososira* and *Nitrosococcus* were the dominant bacterial groups in water as well as surface sediments. Further, comparisons of the five genera in the two environments showed that all their abundances were significantly different in water versus sediments ($p < 0.05$). *Nitrosospira* and *Nitrosococcus* were more abundant in sediments (the relative abundances of them were 44.13% and 41.79% respectively) than in water (40.64% and 29.05%), while *Nitrosomonas*, *Proteobacteria_unclassified*, and *Nitrosomonadaceae_unclassified* were more abundant in water (15.93%, 8.60% and 5.77%) than in sediments (6.98%, 4.53% and 1.55%).

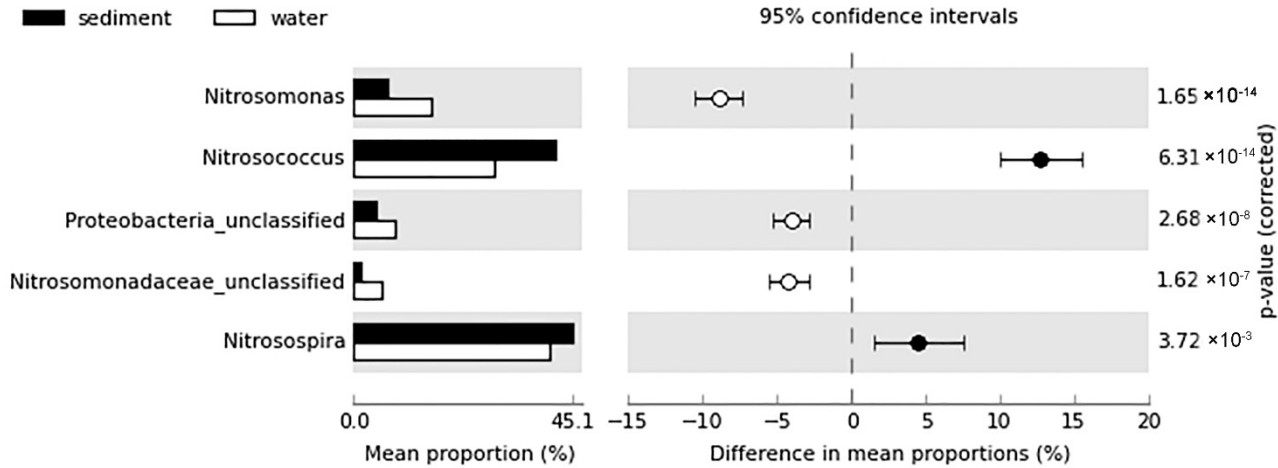

**Figure 1.** The abundance comparisons ammonia-oxidizing bacterial community compositions in genus level between water and surface sediment.

Further, we performed a one-way ANOVA on the abundances of the five genera between different months and ponds. The results showed that monthly changes, rather than pond differences, could cause significant variances in the abundances of all five genera in water ($p < 0.05$) (Figure 2A,B). Meanwhile, monthly changes and pond differences could all cause significant variances in the abundances of all five genera in sediment ($p < 0.05$) (Figure 2C,D).

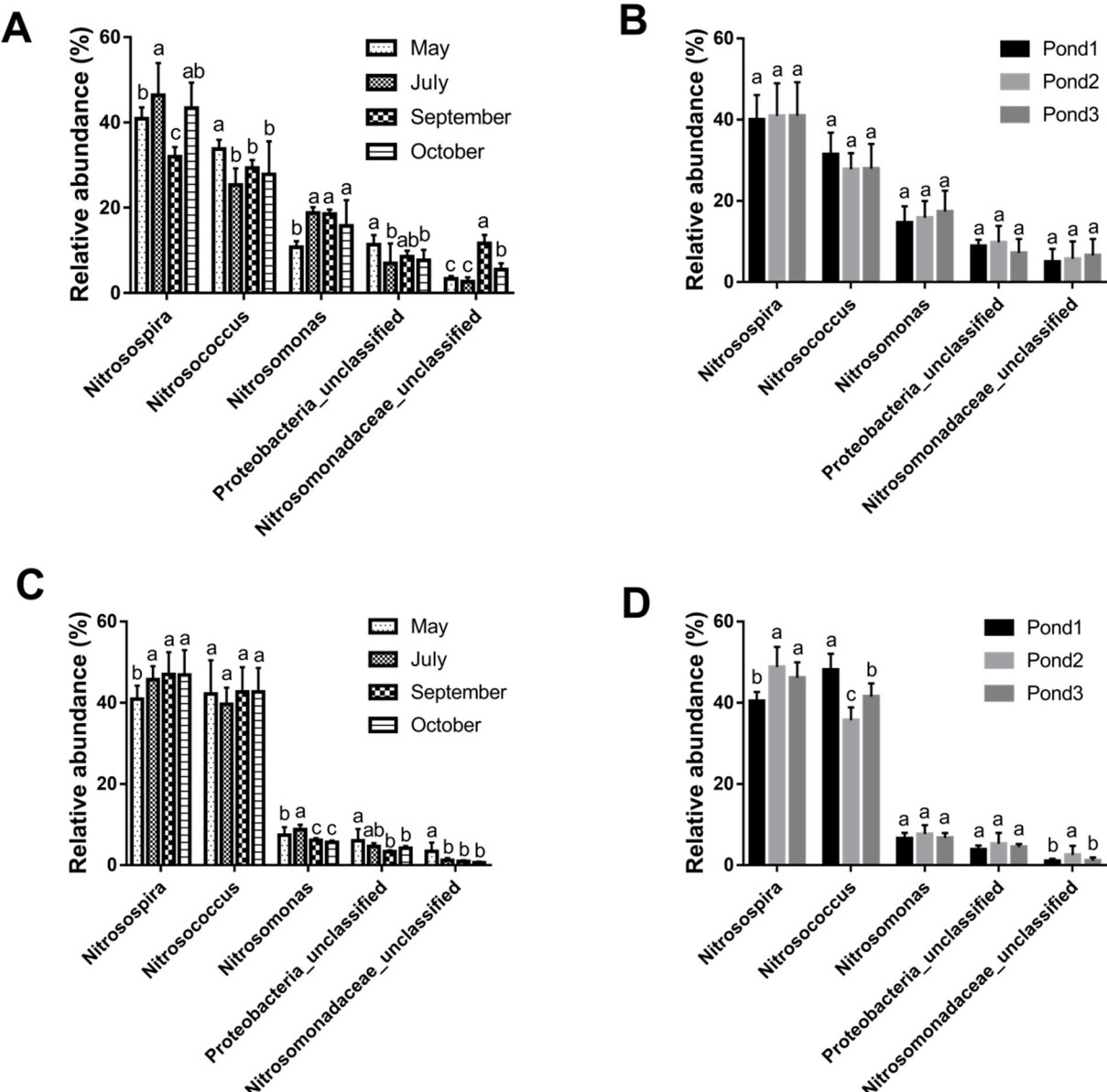

**Figure 2.** The abundance variations of ammonia-oxidizing bacteria in genus level between months changes and pond differences. (**A,B**) represent the abundance variations of the five genera in water with month changes and pond differences, respectively; (**C,D**) represent the abundance changes in the five genera in sediment with month changes and pond differences, respectively. The different letters above the columns mean that there are significant differences between groups at 95% level, "ab" represents that this group has no significant differences with either the group of "a" or "b".

In addition, correlation analyses were performed between the abundances of the three most abundant genera and physio–chemical parameters, and the results are listed in Table 1. The results of the water samples are as follows: only the pH was significantly and positively related to the relative abundance of *Nitrosospira*, while nitrite and nitrate were significantly and negatively related to it. That means that the abundance of *Nitrosospira* increased with the increase in pH value (when the pH is in the neutral to weakly alkaline range), while, as the ammonia-oxidation products, nitrite and nitrate presented an inhibition effect on the abundance of *Nitrosospira*. Nitrite, sulphate and DO were significantly and positively related to the relative abundance of *Nitrosococcus*, while TP, ammonia, and pH were negatively related to it. This indicates that *Nitrosococcus* can significantly reduce the concentration of ammonia and increase the concentration of nitrite under aerobic conditions.

TP, ammonia, and pH were significantly and positively related to the relative abundance of *Nitrosomonas*, while only nitrite was significantly and negatively related to it. This indicates that ammonia promoted the ammonia oxidation reaction based on the existence of *Nitrosomonas* and made the concentration of nitrite increase significantly. In sediments, the relative abundance of *Nitrosospira* was negatively related to TN; the relative abundance of *Nitrosococcus* were positively related to TN, TP, and OM; the relative abundance of *Nitrosomonas* were negatively related to TN, TP, and OM. Moreover, due to the fact that the two most abundant genera, *Nitrosospira* and *Nitrosococcus*, could be affected by pond differences, correlation analyses were also performed between these genera and stock densities (STD) and feed coefficients (FC). The results showed that (Table 1) the abundance of *Nitrosospira* was significantly and positively related to STD and negatively related to FC. Meanwhile, the abundance of *Nitrosococcus* was significantly and positively related to FC.

**Table 1.** Spearman correlation analysis on compositions, richness, and diversity of ammonia-oxidizing bacterial communities and physio–chemical parameters.

| Sources | Parameters | *Nitrosospira* | *Nitrosococcus* | *Nitrosomonas* | Ace | Shannon |
|---|---|---|---|---|---|---|
| water | TP | −0.246 | −0.562 * | 0.730 * | −0.071 | 0.078 |
| | TN | 0.011 | 0.266 | −0.229 | −0.092 | −0.157 |
| | Ammonia | −0.085 | −0.445 * | 0.527 * | 0.122 | −0.087 |
| | Nitrite | −0.393 * | 0.658 * | −0.576 * | 0.012 | −0.330 * |
| | Sulfate | −0.142 | 0.502 * | −0.391 | 0.334 * | −0.460 * |
| | Nitrate | −0.427 * | 0.029 | 0.139 | 0.151 | 0.561 * |
| sediment | TOC | −0.232 | 0.048 | 0.097 | −0.189 | 0.012 |
| | pH | 0.339 * | −0.561 * | 0.369 * | 0.287 | 0.285 |
| | DO | −0.158 | 0.503 * | −0.305 | 0.043 | −0.003 |
| | SD | −0.023 | −0.009 | 0.085 | 0.018 | −0.353 * |
| | FC | 0.265 | −0.016 | −0.154 | −0.056 | 0.000 |
| | STD | −0.275 | 0.180 | 0.239 | 0.033 | −0.206 |
| | sTP | −0.118 | 0.363 * | −0.459 * | 0.064 | −0.129 |
| | sTN | −0.547 * | 0.531 * | −0.407 * | −0.164 | −0.309 |
| | OM | −0.255 | 0.474 * | −0.665 * | 0.123 | −0.327 |
| | STD | 0.524 * | −0.436 | 0.072 | −0.043 | −0.400 * |
| | FC | −0.694 * | 0.851 * | −0.193 | 0.115 | 0.048 |

* Represents *p* < 0.05.

### 3.2. Richness, Diversity of Ammonia-Oxidizing Bacteria and Influencing Factors

Species richness is generally defined as the number of different species, while species diversity is formed from species richness by further classifying the species by some attribute, such as abundance, size, or ecological role. In the present study, the Ace index and the Shannon index were used to define the ammonia-oxidizing bacterial species richness and diversity, respectively. The results showed that they were all affected by monthly variations. The richness of the ammonia-oxidizing bacteria was significantly higher in the later stage of the cultivation cycle (September and October) (Figure 3A), and the diversity (Figure 3B) of the ammonia-oxidizing bacteria was higher in May and September and lower in July and October. Meanwhile, between the three different ponds, only the Ace index value, rather than the Shannon index value, and the abundance of the ammonia-oxidizing bacterial community appeared to be significantly different. In the sediment samples, the ammonia-oxidizing bacterial species richness (Figure 3C) could not be affected by neither the monthly variances nor pond differences. Meanwhile, the ammonia-oxidizing bacterial species' diversity (Figure 3D) was the exact opposite, as it could be affected by both monthly variances and pond differences. It showed the highest value in July, followed by May; the order of the values from high to low was pond 2 > pond 1 > pond 3. The number of reads and OTUs and the values of the Ace and Shannon indices of each sample are listed in Table S1 (water) and Table S2 (sediment).

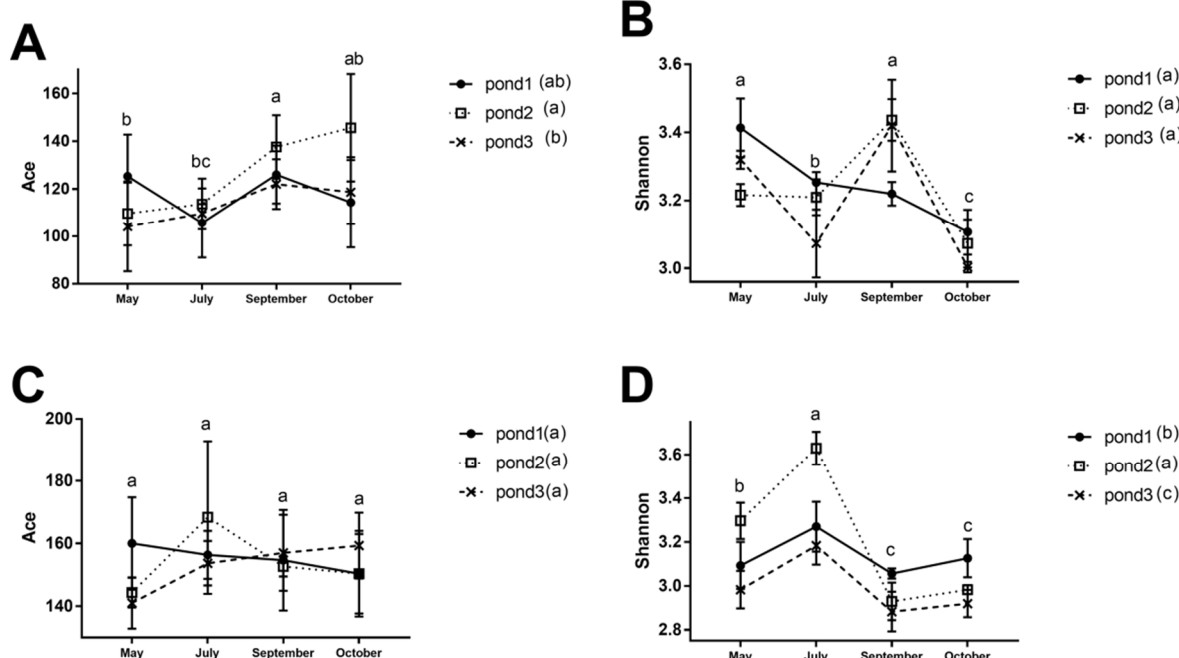

**Figure 3.** Richness (the Ace index) and diversity (the Shannon index) of ammonia-oxidizing bacterial communities in water (**A,B**) and surface sediment (**C,D**) of tilapia ponds. Higher values of Ace index and Shannon index represent higher richness and diversity, respectively. The different letters above the columns mean that there are significant differences between groups at 95% level, "bc" represents that this group has no significant differences with either the group of "b" or "c".

Correlation analyses were also performed between the Ace index values, the Shannon index values, and the physio–chemical parameters (Table 1). The results show that both richness and diversity could not be significantly affected by the three parameters (total phosphorus, total nitrogen, and organic matter) in sediments. Meanwhile, in water, the Ace index appeared to have significantly positive correlations with sulphate ($p < 0.05$), and the Shannon index appeared to have significantly positive correlations with nitrate as well as significantly negative correlations with nitrite, sulphate, and SD.

Correlation analyses also showed that the diversity of ammonia-oxidizing bacteria in the surface sediments of tilapia ponds was negatively related to STD (Table 1).

### 3.3. Ordination Analysis and the Environmental Interpretations

In order to investigate the variations of belt diversity of ammonia-oxidizing bacterial communities, the nonmetric multidimensional scaling (NMDS) method based on the Bray–Curtis similarity was used. The NMDS diagrams of the 36 water samples and 36 sediment samples are listed in Figure 4A,B, respectively. The results show that both the water samples and the sediment samples were separated into three groups, and the ordination basis of the water samples (Figure 4A) and sediment samples (Figure 4B) were months and ponds, respectively. This may suggest that the ammonia-oxidizing bacterial communities in water were more sensitive to monthly changes than to pond differentiations, while the ammonia-oxidizing bacterial communities in sediments were more sensitive to pond differentiations than monthly changes.

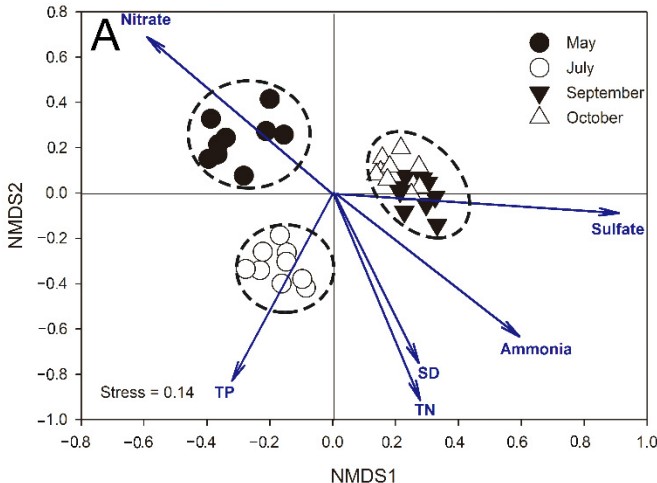
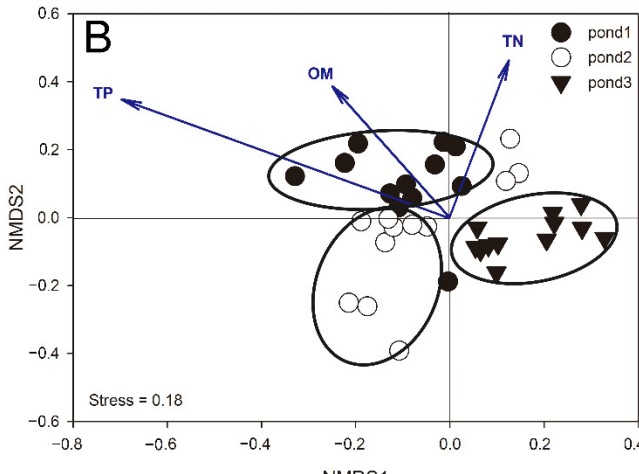

**Figure 4.** NMDS analysis of ammonia-oxidizing bacterial communities of water (**A**) and surface sediment (**B**) samples of the three ponds and their environmental interpretations.

Moreover, in order to understand whether the ordinations were related to the changes in the physico-chemical parameters, we performed an interpretation of the NMDS ordinations of the ammonia-oxidizing bacterial communities with the environmental variables mentioned above (Table 2). The results showed that TP, TN, ammonia nitrite, sulphate, nitrate, TOC, DO, SD, and ORP in water were all significantly related to the NMDS ordinations ($p < 0.05$), and TP, TN, and OM in sediments were all significantly related to the NMDS ordinations ($p < 0.05$). Afterwards, we overlaid the significantly related variables ($p < 0.05$) in water and surface sediments onto their NMDS ordination diagrams, respectively. The results are shown in Figure 4A,B. From Figure 4A (in water), we observed that nitrate was more related to the ammonia-oxidizing bacterial community ordinations in May, TP and TN were more related to such ordinations in July, and sulphate and ammonia in September and October. Further, from Figure 4B, we can observe that TP, TN, and OM were all more related to the ammonia-oxidizing bacterial community ordinations in pond 1.

**Table 2.** Tests of correlation significance on environmental vectors and NMDS ordinations of ammonia-oxidizing bacterial communities in water and sediment.

| Source | Environmental Vectors | $R^2$ | Pr |
|---|---|---|---|
| | TP | 0.796 | 0.001 * |
| | TN | 0.876 | 0.001 * |
| | Ammonia | 0.762 | 0.001 * |
| | Nitrite | 0.399 | 0.002 * |
| | Sulfate | 0.794 | 0.001 * |
| Water | Nitrate | 0.822 | 0.001 * |
| | TOC | 0.296 | 0.003 * |
| | pH | 0.121 | 0.118 |
| | DO | 0.306 | 0.003 * |
| | SD | 0.642 | 0.001 * |
| | ORP | 0.372 | 0.003 * |
| | sTP | 0.608 | 0.001 * |
| Sediment | sTN | 0.231 | 0.019 * |
| | OM | 0.214 | 0.017 * |

* Represents $p < 0.05$.

## 4. Discussion

The present study firstly described the ammonia-oxidizing bacterial communities of both water and surface sediments of tilapia aquaculture ponds, and investigated their

responses to monthly variations, pond differences, and changes in the physico-chemical parameters.

### 4.1. Ammonia-Oxidizing Bacterial Community Compositions in Water and Surface Sediments and Influencing Factors

The ammonia-oxidizing bacterial community compositions at the genus level and the abundant genera between water and surface sediments were all the same. This is similar to that in other natural water ecosystems [20,21]. This indicates a close connection of the ammonia-oxidizing bacterial communities between the two habitats. This connection is based on the similarity of these two habitats and the finite physical distance. The relationship between the physicochemical characteristics of the habitats, the functional processes therein, and the microbial community are often closely interwoven [22]. In addition, the promoting effect of biological movements can also promote the communication of microbes between two habitats. On the other hand, all five genera showed significantly different abundances between water and surface sediments. This may indicate that the environmental factors of these two habitats caused different degrees of functional redundancy with respect to ammonia oxidization [23]. Further, of all the five genera, *Nitrosomonas*, *Proteobacteria_unclassified*, and *Nitrosomonadaceae_unclassified* appeared the least abundant in water, although water may be their main habitat. However, the case for *Nitrosospira* and *Nitrosococcus* was the exact reverse.

In water, the abundances of all five genera were affected by monthly variations rather than by pond differences. This indicates that different cultivation practices, including differences in stocking densities, differences in feed input, etc., could not affect the five ammonia-oxidizing bacterial genera. However, the monthly variations were the main factor causing the abundance changes. Meanwhile, in sediment, the abundances of the two most abundant genera, *Nitrosospira* and *Nitrosococcus*, were hardly affected by monthly variations. They were, however, affected by pond differences. Further, the pond differences were the main influencing factors. As such, the pond differences were closely related to the cultivation management (stocking density and feed coefficients). In addition, lower abundance of *Nitrosopira* and higher abundance of *Nitrosococus* seemed to be a characteristic of the lower stocking density of tilapia.

In the present study, the abundances of *Nitrosospira* and *Nitrosomonas* were all negatively correlated with nitrite in water. This may be related to the fact that these two betaproteobacterial AOB can not only oxidize the ammonia to nitrite, but also carry out nitrifier denitrification under aerobic conditions [24]. In intensive aquaculture ponds, owing to the continuous feed input and fish respiration, the pond water is often in a state of oxygen deficiency. This may supply suitable dissolved oxygen conditions for nitrifier denitrification. Nitrite has also been shown to be toxic to *Nitrosomonas* [25] and to influence the population structure of *Nitrospira* [26]. The abundance of *Nitrosococcus*, to the contrary, was in positive correlation with nitrite. Either way, the results mentioned above could not reflect their decisive roles in nitrite concentrations since ammonia-oxidizing archaea (AOA) also played important roles in the process of ammonia oxidation. In sediment, the abundance of *Nitrosococcus* was positively correlative with TP, TN, and OM. This suggests that *Nitrosococcus_uncultured_bacterium* (the most abundant AOB species in sediment) may be the facultative autotrophic bacteria, which implicates that it could also utilize organic matters as a carbon source. Meanwhile, the abundance of *Nitrosospira* was negatively correlative with TN. Thus, TN in sediment may be a potential factor to be used for adjusting the ratio of these two dominant genera.

### 4.2. Diversity of Ammonia-Oxidizing Bacterial Community in Water and Surface Sediments and Influencing Factors

Studies have shown that spatial and temporal factors could all affect the diversity of ammonia-oxidizing bacteria in aquatic ecosystems [27–31]. The present study is not an exception. Moreover, ammonia-oxidizing bacterial diversity in water and sediment of tilapia ponds showed different response patterns to spatial and temporal variations. As

mentioned above, pond differences could only affect the diversity of ammonia-oxidizing bacterial community in sediments rather than that in water, which may suggest that the ammonia-oxidizing bacterial community in sediments was more sensitive to the differences of the culture operations. As far as we know, the ammonia in the aquaculture pond was generated mainly from sediments and the concentrations of ammonia increased with the sediment depth [32]. Further, the ammonia-oxidizing bacteria was mainly distributed in surface sediments, rather than in deep layers of the sediment [8]. Thus, concentrations of ammonia in pore water increased over time as fish grew and feeding rates increased, which may explain why the ammonia-oxidizing bacteria in surface sediments responded first. Meanwhile, concentrations of ammonia in water that diffused from the sediments were not high enough to make ammonia-oxidizing bacteria in water respond in time. The physico-chemical conditions in the water, by contrast, were more easily affected by seasonal variances, which may be the reason why diversity of the ammonia-oxidizing bacteria in water could be affected by monthly variances. Beta diversity reflected by an NMDS analysis further supported the findings that the ammonia-oxidizing bacterial community in water and sediments was mainly affected by monthly variances and pond differences, respectively.

In addition to spatial and temporal factors, many environmental parameters, such as salinity [33], DO [34], temperature [35], pH [36], etc., have been proven to be able to influence the ammonia-oxidizing bacterial diversity. The present study shows that, in water, most of the environmental parameters were related to the variations of the beta-diversity of the ammonia-oxidizing bacterial diversity, which may indicate that the variations of the ammonia-oxidizing bacterial communities with the monthly changes were all related to these environmental parameters. TN, nitrate, TP, sulphate, and ammonia were more related. Meanwhile, only nitrite, sulphate, nitrate, and SD were significantly related to changes in the alpha-diversity (Shannon index). Among them, nitrate was positively correlated with the values of the Shannon indices, and nitrite, sulphate, and SD were negatively correlated with the values of the Shannon indices. Nitrate was negatively correlated with *Nitrosospira* abundance, which may also be attributed to the nitrifier denitrifications carried out by *Nitrosospira* under aerobic conditions. Further, this may be the reason that nitrate was positively correlated with the Shannon indices of ammonia-oxidizing bacteria. The abundance reduction of the most abundant genus, *Nitrosospira*, together with the abundance increase in the less abundant genera, made the Shannon indices of ammonia-oxidizing bacteria increase. Moreover, the nitrite being toxic to some of the ammonia-oxidizing genera may be the cause of influencing the Shannon indices of the ammonia-oxidizing bacteria. In sediments, the beta-diversity of ammonia-oxidizing bacteria could all be influenced by TP, TN, and OM. In particular, TP has a relatively higher correlation coefficient, which may indicate that the different contents of TP was one of the main factors causing the differences in the ammonia-oxidizing bacterial communities between the different ponds.

Combining the above two aspects (Sections 4.1 and 4.2), the adjusting of the TN and TP levels in sediment may be an effective way to affect the ammonia-oxidizing bacterial communities in sediment. This can also be a way to control the ammonia release from the sediment into the tilapia aquaculture water. Some further research based on this can help to develop some methods for water quality control.

## 5. Conclusions

In summary, the ammonia-oxidizing bacterial communities in water and surface sediments of tilapia ponds were significantly different, while their compositions at the genus level were the same, being as follows: *Nitrosospira*, *Nitrosococcus*, *Nitrosomonas*, *Proteobacteria_unclassified*, and *Nitrosomonadaceae_unclassified*. Nevertheless, both the abundances and diversity of the ammonia-oxidizing bacteria communities were mainly affected by monthly variations in water and by pond differences in sediment. Moreover, TN, nitrate, TP, and sulphate were the main factors influencing the ammonia-oxidizing bacterial communities in water, while TN and TP were the main influencing factors in sediments. Further, the

parameter changes, especially those caused by pond differences, were closely related to the cultivation management (stocking density and feed coefficients).

**Supplementary Materials:** The following supporting information can be downloaded at: https://www.mdpi.com/article/10.3390/app12073438/s1, Figure S1: Venn diagram showing the unique and shared OTUs in libraries representing samples of water (W) and sediment (S), Table S1: Richness and diversity of ammonia-oxidizing bacterial communities in each water samples, Table S2: Richness and diversity of ammonia-oxidizing bacterial communities in each sediment samples.

**Author Contributions:** Conceptualization, L.F., S.M. and J.C.; methodology, L.F. and J.C.; sample acquisition L.Q. and C.S.; chemical analyses G.H., L.Q., C.S. and D.L.; writing—original draft preparation, L.F.; writing—review and editing, L.F. and S.M. All authors have read and agreed to the published version of the manuscript.

**Funding:** We thank for supporting our research by China Agriculture Research System of MOF and MARA (Grant CARS-46).

**Institutional Review Board Statement:** Not applicable.

**Informed Consent Statement:** Not applicable.

**Data Availability Statement:** The data presented in this study are available on request from the corresponding authors.

**Conflicts of Interest:** The authors declare no conflict of interest.

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
