# Peer review of "Ammonia-Oxidizing Bacterial Communities in Tilapia Pond Systems and the Influencing Factors"

_applsci, doi:10.3390/app12073438_

Round 1

Reviewer 1 Report

Dear Authors,

An important subject of research is the studies on influencing factors on bacterial communities in natural and artificial systems. The presented manuscript is very interesting. Your studies are present interesting data on how the seasonal variations, cultivation management, and physicochemical parameters may influence bacterial communities in the natural environment. Your studies may implication 

The manuscript has the following problems:
1.    Paragraph 2.4 is only one sentence. I think maybe it will be better to add this sentence to paragraph 2.3 and change its name to bioinformatics and statistical analysis.
2.    The p-values should be added to table 1. 
3.    Table 1 has missing information on what “*” means. I think that short names should be expanded in the caption of Tables 1, and 2, even if they were previously explained. This will facilitate the reception of tables 1, and 2 for readers.
4.    The citation of table 2 is missing in the text of the manuscript. 

Could you explain to me why you used the Spearman correlation? What was the distribution of the results? 

Author Response

Response to Reviewer 1 Comments

Point 1: Paragraph 2.4 is only one sentence. I think maybe it will be better to add this sentence to paragraph 2.3 and change its name to bioinformatics and statistical analysis.

Response 1: We added this sentence to paragraph 2.3 and change its name to bioinformatics and statistical analysis according to the comments.

Point 2: The p-values should be added to table 1.

Response 2: We added the denotes that * presents that p<0.05.

Point 3: Table 1 has missing information on what “*” means. I think that short names should be expanded in the caption of Tables 1, and 2, even if they were previously explained. This will facilitate the reception of tables 1, and 2 for readers.

Response 3: We added the denotes that * presents that p<0.05 and we added the detail of all the abbreviations

Point 4: The citation of table 2 is missing in the text of the manuscript.

Could you explain to me why you used the Spearman correlation? What was the distribution of the results?

Response 4: We added the missing citation of table2. We used the spearman correlation because some parameters are not normally distributed.

Reviewer 2 Report

This paper is an investigation on ammonia-oxidizing bacterial communities in water and surface sediments of three tilapia ponds and their relationship with differences in the ponds, monthly variations in water, and physico-chemical parameters. The article is well written and structured. The English grammar and spellings are adequate. 

According to the final results of this research, ammonia-oxidizing bacterial communities in water and surface sediments of tilapia ponds were significantly influenced by sediments and nutrients. Thus, it would be a good idea to indicate how the results of this research can be useful to the bacterial spread in Tilapia ponds. In other words, what would the authors suggest to prevent or reduce the risks to this matter?

Author Response

Response to Reviewer 2 Comments

Point 1: According to the final results of this research, ammonia-oxidizing bacterial communities in water and surface sediments of tilapia ponds were significantly influenced by sediments and nutrients. Thus, it would be a good idea to indicate how the results of this research can be useful to the bacterial spread in Tilapia ponds. In other words, what would the authors suggest to prevent or reduce the risks to this matter?

Response 1: We added this information according to the comments.

Reviewer 3 Report

Summary:

The presented manuscript explores the role of different environmental conditions on the populations of nitrifying bacteria in tilapia ponds. It identifies the main phylogenetic groups of ammonia-oxidizing bacteria present in the examined ponds and presents the correlations between their share in the samples and the environmental factors. The article presents interesting metagenomic data followed by a discussion in the light of other research dealing with nitrifying bacteria.

General comments:

The presented manuscript presents an extensive in silico metagenomic research on ammonia-oxidizing bacteria from tilapia ponds. However, multiple major and minor flaws can be identified in the article as well as in the experimental design.

Firstly, I’d like to highlight that textual layer of the manuscript should be well revised (and perhaps proofread). In addition to some English language issues, there are many other types of errors, one of which can be even seen in the title (“...their the influencing factors”). One major flaw is the lack of explanations for many of the abbreviations used throughout the manuscript. Due to this problem, the research cannot be fully understood. An example of this problem is that the most important factors of the research, the environmental factors which are supposed to influence microbial communities, are not explained in the manuscript, only given by the abbreviation (“TP”, “OM” “TD” and others). Overall, the writing of manuscript is unclear and inconsistent. Other than this, the formatting of the citations is also not consistent and should be revised as well (missing journal names, occasional journal name abbreviations, inconsistent inclusion of DOI numbers).

Secondly, referring to the experimental design, the article does not clearly state the hypothesis nor the importance of the study. The introduction is easy to follow, however it lacks the central part, which is what is the real aim of the study. From the context it looks like the research is highly preliminary, however some future application should be at least discussed if the authors wish to publish the manuscript in a journal focused on applied research. My questions while reading the article were mostly about the merit of the study. The authors do not state any benefits from the obtained information. How can the results affect the tilapia farming? What is the potential of using this new knowledge?

The methods section is very brief. My biggest problem was that I was not able to understand what the environmental factors tested are, this mostly was due to the unexplained abbreviations. However, this would be a little easier to understand if the methodology of measurements of these factors in the samples was given in the manuscript. Authors should also elaborate on the statistical analysis since it is an extensive part of the study.

The results section only states the statistical analysis results. However, there is no information on how exactly do the pH or other environmental factors affect the nitrifying microbiome of the ponds. I suggest deeper analyses that would show which factors promote the abundance of which groups. The sole fact that the environmental factors do affect various bacterial groups presence does not provide enough novelty to admit this study as an advance in the field of microbiology.

The discussion section presents the newly obtained knowledge with comparison to some previous studies. Several interesting conclusions can be found, such as the paragraph in lines 277-285. Nevertheless, there is no broader context presented for the given research which sparks my concern about this study being published in a journal like Applied Sciences, which gathers research with application potential. The relevance of the manuscript to Applied Sciences journal is questionable.

Detailed comments:

Line 64: The authors state that they intend to investigate the mechanism of the bacterial responses to environmental factors, however the presented experimental data do not attempt to explain this mechanism. The experiments included in the study mostly answer the question whether the environment affects the bacteria at all and this is not explaining mechanisms.

Figure 1: This figure could be successfully joined with Figure 2. (as an A and B parts), moved to supplementary data or even deleted from the manuscript since it seems a little unnecessary. This part is well explained in the text and it doesn’t need a simplification in a graph.

Line 159: Which post-hoc test was used as a complement to the one-way ANOVA? The significant differences are stated in the figure, but there is no explanation of how they were calculated.

Line 180: Could the authors explain why the feed coefficient was only used for tests on sediment samples, but not in the water samples? It seems relevant to water samples as well.

Table 1: Unfortunately, it is impossible to understand the Table (and the main text) since the abbreviations were not explained anywhere in the manuscript. This is perhaps my biggest problem with this study, as it cannot be comprehended without knowing what were the environmental factors measured.

Line 196: The concepts of the Ace index and Shannon index are introduced very briefly. The ideas they represent, namely “richness” and “diversity”, should be clearly explained in the context of this research. What does it mean that we have more “richness” or more “diversity” in a sample?

Line 224: The aim of the NMDS analysis is not clearly stated. What was the hypothesis and what did the authors intend to observe provided with the NMDS analysis results?

Author Response

Response to Reviewer 3 Comments

Point 1: Firstly, I’d like to highlight that textual layer of the manuscript should be well revised (and perhaps proofread). In addition to some English language issues, there are many other types of errors, one of which can be even seen in the title (“...their the influencing factors”). One major flaw is the lack of explanations for many of the abbreviations used throughout the manuscript. Due to this problem, the research cannot be fully understood. An example of this problem is that the most important factors of the research, the environmental factors which are supposed to influence microbial communities, are not explained in the manuscript, only given by the abbreviation (“TP”, “OM” “TD” and others). Overall, the writing of manuscript is unclear and inconsistent. Other than this, the formatting of the citations is also not consistent and should be revised as well (missing journal names, occasional journal name abbreviations, inconsistent inclusion of DOI numbers).

Response 1: We revised these errors according to the comments.

Point 2: Secondly, referring to the experimental design, the article does not clearly state the hypothesis nor the importance of the study. The introduction is easy to follow, however it lacks the central part, which is what is the real aim of the study. From the context it looks like the research is highly preliminary, however some future application should be at least discussed if the authors wish to publish the manuscript in a journal focused on applied research. My questions while reading the article were mostly about the merit of the study. The authors do not state any benefits from the obtained information. How can the results affect the tilapia farming? What is the potential of using this new knowledge?

Response 2: We stated the hypothesis and the importance of this study in introdution section. And we stated how the obtained inforamtion help the tilapia in the discussion section.

Point 3: The methods section is very brief. My biggest problem was that I was not able to understand what the environmental factors tested are, this mostly was due to the unexplained abbreviations. However, this would be a little easier to understand if the methodology of measurements of these factors in the samples was given in the manuscript. Authors should also elaborate on the statistical analysis since it is an extensive part of the study.

Response 3: We added this sentence to paragraph 2.3 and change its name to bioinformatics and statistical analysis according to the comments.

Point 4: The results section only states the statistical analysis results. However, there is no information on how exactly do the pH or other environmental factors affect the nitrifying microbiome of the ponds. I suggest deeper analyses that would show which factors promote the abundance of which groups. The sole fact that the environmental factors do affect various bacterial groups presence does not provide enough novelty to admit this study as an advance in the field of microbiology.

Response 4: We added this sentence to paragraph 2.3 and change its name to bioinformatics and statistical analysis according to the comments.

Point 5: The discussion section presents the newly obtained knowledge with comparison to some previous studies. Several interesting conclusions can be found, such as the paragraph in lines 277-285. Nevertheless, there is no broader context presented for the given research which sparks my concern about this study being published in a journal like Applied Sciences, which gathers research with application potential. The relevance of the manuscript to Applied Sciences journal is questionable.

Response 5: We added this sentence to paragraph 2.3 and change its name to bioinformatics and statistical analysis according to the comments.

Point 6: Line 64: The authors state that they intend to investigate the mechanism of the bacterial responses to environmental factors, however the presented experimental data do not attempt to explain this mechanism. The experiments included in the study mostly answer the question whether the environment affects the bacteria at all and this is not explaining mechanisms.

Response 6: We revised the context that not used the word of mechanism.

Point 7: Figure 1: This figure could be successfully joined with Figure 2. (as an A and B parts), moved to supplementary data or even deleted from the manuscript since it seems a little unnecessary. This part is well explained in the text and it doesn’t need a simplification in a graph.

Response 7: We moved Figure1 to supplementary data according to the comments.

Point 8: Line 159: Which post-hoc test was used as a complement to the one-way ANOVA? The significant differences are stated in the figure, but there is no explanation of how they were calculated.

Response 8: We added the information about the post-hoc test method in the materials and method section.

Point 9: Line 180: Could the authors explain why the feed coefficient was only used for tests on sediment samples, but not in the water samples? It seems relevant to water samples as well.

Response 9: We added this information.

Point 10: Table 1: Unfortunately, it is impossible to understand the Table (and the main text) since the abbreviations were not explained anywhere in the manuscript. This is perhaps my biggest problem with this study, as it cannot be comprehended without knowing what were the environmental factors measured..

Response 10: We added 2.4 in materials and method section to tell all the details of environmental factors.

Point 11: Line 196: The concepts of the Ace index and Shannon index are introduced very briefly. The ideas they represent, namely “richness” and “diversity”, should be clearly explained in the context of this research. What does it mean that we have more “richness” or more “diversity” in a sample?

Response 11: We did some explainations on richness and diversity in this section.

Point 12: Line 224: The aim of the NMDS analysis is not clearly stated. What was the hypothesis and what did the authors intend to observe provided with the NMDS analysis results?

Response 12: We added this information before this section.

Reviewer 4 Report

Acceptable 

Author Response

Point1: Acceptable 

Response1:Thanks

Round 2

Reviewer 3 Report

Reviewer's comments to: "Response to Reviewer 3 Comments"

Point 1: Firstly, I’d like to highlight that textual layer of the manuscript should be well revised (and perhaps proofread). In addition to some English language issues, there are many other types of errors, one of which can be even seen in the title (“...their the influencing factors”). One major flaw is the lack of explanations for many of the abbreviations used throughout the manuscript. Due to this problem, the research cannot be fully understood. An example of this problem is that the most important factors of the research, the environmental factors which are supposed to influence microbial communities, are not explained in the manuscript, only given by the abbreviation (“TP”, “OM” “TD” and others). Overall, the writing of manuscript is unclear and inconsistent. Other than this, the formatting of the citations is also not consistent and should be revised as well (missing journal names, occasional journal name abbreviations, inconsistent inclusion of DOI numbers).

Response 1: We revised these errors according to the comments.

Reviewer response: Although some parts were actually revised and corrected, the authors only focused on the parts that were mentioned in the review. However, as also mentioned, the overall quality of language in the manuscript is poor. I advise an extensive language editing in order to provide the readers with clear communication of the study’s results.

Point 2: Secondly, referring to the experimental design, the article does not clearly state the hypothesis nor the importance of the study. The introduction is easy to follow, however it lacks the central part, which is what is the real aim of the study. From the context it looks like the research is highly preliminary, however some future application should be at least discussed if the authors wish to publish the manuscript in a journal focused on applied research. My questions while reading the article were mostly about the merit of the study. The authors do not state any benefits from the obtained information. How can the results affect the tilapia farming? What is the potential of using this new knowledge?

Response 2: We stated the hypothesis and the importance of this study in introdution section. And we stated how the obtained inforamtion help the tilapia in the discussion section.

Reviewer response: This part is now clear.

Point 3: The methods section is very brief. My biggest problem was that I was not able to understand what the environmental factors tested are, this mostly was due to the unexplained abbreviations. However, this would be a little easier to understand if the methodology of measurements of these factors in the samples was given in the manuscript. Authors should also elaborate on the statistical analysis since it is an extensive part of the study.

Response 3: We added this sentence to paragraph 2.3 and change its name to bioinformatics and statistical analysis according to the comments.

Reviewer response: The methods section is still short and does not explain everything. I appreciate the new paragraph on the measurments of variable factors, all it does however is sending the reader to another publication. My opinion on this is that we should provide at least a brief description of every method that is used in the study. The article should be as standalone as possible and this is not the case here.

Point 4: The results section only states the statistical analysis results. However, there is no information on how exactly do the pH or other environmental factors affect the nitrifying microbiome of the ponds. I suggest deeper analyses that would show which factors promote the abundance of which groups. The sole fact that the environmental factors do affect various bacterial groups presence does not provide enough novelty to admit this study as an advance in the field of microbiology.

Response 4: We added this sentence to paragraph 2.3 and change its name to bioinformatics and statistical analysis according to the comments.

Reviewer response: The authors response is not answering my comment. Please provide a suitable reply.

Point 5: The discussion section presents the newly obtained knowledge with comparison to some previous studies. Several interesting conclusions can be found, such as the paragraph in lines 277-285. Nevertheless, there is no broader context presented for the given research which sparks my concern about this study being published in a journal like Applied Sciences, which gathers research with application potential. The relevance of the manuscript to Applied Sciences journal is questionable.

Response 5: We added this sentence to paragraph 2.3 and change its name to bioinformatics and statistical analysis according to the comments.

Reviewer response: Although some changes have been introduced in the Discussion section, the authors response is not answering my comment. Please provide a suitable reply.

Point 6: Line 64: The authors state that they intend to investigate the mechanism of the bacterial responses to environmental factors, however the presented experimental data do not attempt to explain this mechanism. The experiments included in the study mostly answer the question whether the environment affects the bacteria at all and this is not explaining mechanisms.

Response 6: We revised the context that not used the word of mechanism.

Reviewer response: This point is now resolved and clear

Point 7: Figure 1: This figure could be successfully joined with Figure 2. (as an A and B parts), moved to supplementary data or even deleted from the manuscript since it seems a little unnecessary. This part is well explained in the text and it doesn’t need a simplification in a graph.

Response 7: We moved Figure1 to supplementary data according to the comments.

Reviewer response: This point is now resolved.

Point 8: Line 159: Which post-hoc test was used as a complement to the one-way ANOVA? The significant differences are stated in the figure, but there is no explanation of how they were calculated.

Response 8: We added the information about the post-hoc test method in the materials and method section.

Reviewer response: This point is now resolved and the test name is stated in the methods.

Point 9: Line 180: Could the authors explain why the feed coefficient was only used for tests on sediment samples, but not in the water samples? It seems relevant to water samples as well.

Response 9: We added this information.

Reviewer response: Although the authors state that they changed the manuscript according to the comment, I cannot find this explanation anywhere in the manuscript. Please provide the details of where to find it or correct the manuscript if it hasn’t been corrected with respect to this comment yet.

Point 10: Table 1: Unfortunately, it is impossible to understand the Table (and the main text) since the abbreviations were not explained anywhere in the manuscript. This is perhaps my biggest problem with this study, as it cannot be comprehended without knowing what were the environmental factors measured..

Response 10: We added 2.4 in materials and method section to tell all the details of environmental factors.

Reviewer response: This point is now clear. However, since the authors stated the abbreviations’ explanations in the methods section, the footnote below the table has some redundancies as it duplicates the explanations. I believe explaining an abbreviation once would be enough for the clarity on this matter.

Point 11: Line 196: The concepts of the Ace index and Shannon index are introduced very briefly. The ideas they represent, namely “richness” and “diversity”, should be clearly explained in the context of this research. What does it mean that we have more “richness” or more “diversity” in a sample?

Response 11: We did some explainations on richness and diversity in this section.

Reviewer response: This point was addressed in the manuscript very nicely and is now clear.

Point 12: Line 224: The aim of the NMDS analysis is not clearly stated. What was the hypothesis and what did the authors intend to observe provided with the NMDS analysis results?

Response 12: We added this information before this section.

Reviewer response: This point is now resolved, although I suggest rephrasing the sentence to be more clear.

Author Response

Point 1: Firstly, I’d like to highlight that textual layer of the manuscript should be well revised (and perhaps proofread). In addition to some English language issues, there are many other types of errors, one of which can be even seen in the title (“...their the influencing factors”). One major flaw is the lack of explanations for many of the abbreviations used throughout the manuscript. Due to this problem, the research cannot be fully understood. An example of this problem is that the most important factors of the research, the environmental factors which are supposed to influence microbial communities, are not explained in the manuscript, only given by the abbreviation (“TP”, “OM” “TD” and others). Overall, the writing of manuscript is unclear and inconsistent. Other than this, the formatting of the citations is also not consistent and should be revised as well (missing journal names, occasional journal name abbreviations, inconsistent inclusion of DOI numbers).

Response 1: We revised these errors according to the comments.

Reviewer response: Although some parts were actually revised and corrected, the authors only focused on the parts that were mentioned in the review. However, as also mentioned, the overall quality of language in the manuscript is poor. I advise an extensive language editing in order to provide the readers with clear communication of the study’s results.

Response: The English for our manuscript was improved by .the editage firm.

Point 3: The methods section is very brief. My biggest problem was that I was not able to understand what the environmental factors tested are, this mostly was due to the unexplained abbreviations. However, this would be a little easier to understand if the methodology of measurements of these factors in the samples was given in the manuscript. Authors should also elaborate on the statistical analysis since it is an extensive part of the study.

Response 3: We added this sentence to paragraph 2.3 and change its name to bioinformatics and statistical analysis according to the comments.

Reviewer response: The methods section is still short and does not explain everything. I appreciate the new paragraph on the measurments of variable factors, all it does however is sending the reader to another publication. My opinion on this is that we should provide at least a brief description of every method that is used in the study. The article should be as standalone as possible and this is not the case here.

Response: We added this information according to the comment.

Point 4: The results section only states the statistical analysis results. However, there is no information on how exactly do the pH or other environmental factors affect the nitrifying microbiome of the ponds. I suggest deeper analyses that would show which factors promote the abundance of which groups. The sole fact that the environmental factors do affect various bacterial groups presence does not provide enough novelty to admit this study as an advance in the field of microbiology.

Response 4: We added this sentence to paragraph 2.3 and change its name to bioinformatics and statistical analysis according to the comments.

Reviewer response: The authors response is not answering my comment. Please provide a suitable reply.

Response: We added the information about how the environmental factors and the abundant genera interacted each other.

Point 5: The discussion section presents the newly obtained knowledge with comparison to some previous studies. Several interesting conclusions can be found, such as the paragraph in lines 277-285. Nevertheless, there is no broader context presented for the given research which sparks my concern about this study being published in a journal like Applied Sciences, which gathers research with application potential. The relevance of the manuscript to Applied Sciences journal is questionable.

Response 5: We added this sentence to paragraph 2.3 and change its name to bioinformatics and statistical analysis according to the comments.

Reviewer response: Although some changes have been introduced in the Discussion section, the authors response is not answering my comment. Please provide a suitable reply.

Response: We improved this section according to the comment.

Point 9: Line 180: Could the authors explain why the feed coefficient was only used for tests on sediment samples, but not in the water samples? It seems relevant to water samples as well.

Response 9: We added this information.

Reviewer response: Although the authors state that they changed the manuscript according to the comment, I cannot find this explanation anywhere in the manuscript. Please provide the details of where to find it or correct the manuscript if it hasn’t been corrected with respect to this comment yet.

Response: We added this information to Table.1

Point 10: Table 1: Unfortunately, it is impossible to understand the Table (and the main text) since the abbreviations were not explained anywhere in the manuscript. This is perhaps my biggest problem with this study, as it cannot be comprehended without knowing what were the environmental factors measured..

Response 10: We added 2.4 in materials and method section to tell all the details of environmental factors.

Reviewer response: This point is now clear. However, since the authors stated the abbreviations’ explanations in the methods section, the footnote below the table has some redundancies as it duplicates the explanations. I believe explaining an abbreviation once would be enough for the clarity on this matter.

Response: We removed this sentence from the footnote.
